# Rhizospheric *Bacillus* spp. Exhibit Miticidal Efficacy against *Oligonychus coffeae* (Acari: Tetranychidae) of Tea

**DOI:** 10.3390/microorganisms11112691

**Published:** 2023-11-02

**Authors:** Popy Bora, Sukanya Gogoi, Mukund Vinayak Deshpande, Pankaj Garg, Rana P. Bhuyan, Nilofar Altaf, Nikita Saha, Sapna Mayuri Borah, Mousumi Phukon, Nabajit Tanti, Bishal Saikia, Shenaz Sultana Ahmed, Sanjib Ranjan Borah, Ashish Dutta, Bidyut Kumar Sarmah

**Affiliations:** 1Biocontrol Laboratory, DBT-North East Centre for Agricultural Biotechnology, Jorhat 785013, Indiabidyut.sarmah@aau.ac.in (B.K.S.); 2AAU-Assam Rice Research Institute, Assam Agricultural University, Jorhat 785013, India; sgogoi119@gmail.com (S.G.);; 3National Chemical Laboratory, Pune 411008, India; mvdeshpande1952@gmail.com; 4Department of Chemistry, GLA University, Mathura 281406, India; pankaj.garg@gla.ac.in; 5Department of Tea Husbandry and Technology, Assam Agricultural University, Jorhat 785013, India; 6Department of Entomology, Assam Agricultural University, Jorhat 785013, India; 7Department of Plant Pathology, Assam Agricultural University, Jorhat 785013, India

**Keywords:** *Bacillus* spp., biocontrol, organic tea, secondary metabolites

## Abstract

*Oligonychus coffeae* (Acari: Tetranychidae), popularly known as red spider mite (RSM) is one of the major pests of commercial tea (*Camellia sinensis* (L.) O. Kuntze) plantation world over. Many attempts have been made in the past to control this devastating pest using a variety of microbial bioagents, however, area-wise field success is very limited. We carried out an in vitro study to explore the potential of rhizospheric *Bacillus* spp. (*B. amyloliquefaciens* BAC1, *B. subtilis* LB22, and *B. velezensis* AB22) against *O. coffeae* through adulticidal and ovicidal activity. The 100% adult and egg mortality was observed with bacterial suspension (1 × 10^9^ CFU/mL) by *B. velezensis* AB22, showing the lowest LC_50_ values for both adults and eggs of *O. coffeae*, i.e., 0.28 × 10^5^ and 0.29 × 10^5^, respectively. The study also throws some insights into the underlying mechanism through electron microscopy study and identification of some putative pesticidal metabolites from all the species. The three *Bacillus* species were observed to have four commonly secreted putative bioactive secondary metabolites, brevianamide A, heptadecanoic acid, thiolutin, and versimide responsible for their bio-efficacy against *O. coffeae*. The outcome of our study provides a strong possibility of introducing *Bacillus* spp. as a biological miticide and developing synthetic metabolites mimicking the mechanistic pathway involved in microbial bioefficacy.

## 1. Introduction

Tea, *Camellia sinensis* (L.) O. Kuntze, an evergreen perennial crop of the family Theaceae represents globally one of the oldest organized agricultural industries. Tea made from its tender leaves is the most extensively consumed non-alcoholic beverage throughout the world owing to its multiple health benefits attributable to high antioxidant compounds and polyphenols [1]. The Assam state of India is the single largest contiguous tea growing region in the world, with a production of 0.75 million tons from 0.32 million ha area, thereby, contributing nearly 51% of the global tea basket [2]. The tea plants known as ‘single most forest species’ raised under a monocropping system are subjected to infestation by various pests and diseases, cutting their productive life and deteriorating their quality and quantity, leading to significant damage to the tea industry. Red Spider Mite (RSM), being polyphagous in nature, is one of the major pests of tea [3], causing damage by sucking the sap and lacerating the cells with characteristic reddish-brown marks on the upper surface of mature leaves [4,5], thereby, incurring substantial crop losses ranging between 17% and 46% [4]. Being polyphagous in nature, the pest is reported to cause severe damage to approximately 133 crops in tropical and subtropical regions [6]. Tea growers eventually resort to heavy and frequent use of a number of chemical acaricides. Such intensive, prolonged, and repeated use of synthetic pesticides has many disadvantages, such as non-selective destruction of beneficial microbes, chemical residue-driven soil contamination, and development of pesticide resistance [7,8]. Accumulation of pesticide residues above maximum residue limit in processed tea has resulted in restrictions imposed by tea importers. Such a scenario of tea growing has warranted an immediate necessity to develop an alternative sustainable management option as a green solution without compromising with either green leaf yield or tea quality. Hence, the onus is directed towards exploring microbial pesticides/biopesticides. Different species of *Bacillus* have been widely exploited for disease management in organic production systems [9]. The entomopathogenic bacteria, *Bacillus thuringiensis,* has been used as a popular microbial bioagent option over the last 50 years in tea plantations against lepidopteran pests [10,11,12,13], while other species from the genus *Bacillus* are mostly used as bioagents and growth promoters in an array of crops. Different *Bacillus* spp. have been reported to be highly effective in reducing the black rot severity caused by *Corticium theae* [14] and anthracnose in tea caused by *Colletotrichum theae-sinensis* by 77.30% under glasshouse conditions [15]. A few studies also suggest the entomopathogenic activity of other *Bacillus* spp. owing to its potential to produce a wide range of secondary metabolites, however, such studies are scarce. Further, studies reporting the role of secondary metabolites synthesized by different *Bacillus* species against crop pests need a thorough revisit.

In light of these facts, we investigated the biocontrol efficacy of three different *Bacillus* spp. (*B. amyloliquefaciens* BAC1, *B. subtilis* LB22, and *B. velezensis* AB22) previously isolated from rhizospheric soil of agricultural fields against *O. coffeae* using their adulticidal and ovicidal properties under controlled laboratory conditions. Furthermore, the identification of effective metabolites responsible for their pesticidal activity was also studied to develop a novel approach against mite pests in the organic tea ecosystem. The results of these studies are likely to put forth some new strategies for organic pest management.

## 2. Material and Methods

Experiments were conducted at the Biological Control Laboratory, Department of Plant Pathology, Assam Agricultural University, Jorhat, Assam (India).

### 2.1. Bacterial Strains

Three *Bacillus* spp. isolated from tea rhizosphere with National Centre for Biotechnology Information (NCBI, Bethesda, MD, USA) accessions, *B. amyloliquefaciens* BAC1 (ON392425), *B. subtilis* LB22 (ON386193), and *Bacillus velezensis* AB22 (ON209629) (Appendix A–C) were collected from author’s Biocontrol Laboratory of Assam Agricultural University, Jorhat, Assam (India). All selected strains were maintained in the freezer at −80 °C in 20% glycerol until further use.

The three strains were further submitted to the national repository of the National Bureau of Agriculturally Important Microorganisms (ICAR-NBAIM, Uttar Pradesh, India) with accession numbers, *B*. *amyloliquefaciens* BAC1 (NAIMCC-B-03217), *B. subtilis* LB22 (NAIMCC-B-03226) and *B. velezensis* (NAIMCC-B-03221). All the isolates were subcultured on Nutrient Agar media (NA, Hi-Media, India) with media composition: Peptone (5 gms), HM Peptone B (1.50 gms), Yeast Extract (1.50 gms), NaCl (5.00 gms), Agar (15.00 gms), pH 7.4 and stored at 4 °C for further experiments.

### 2.2. Preparation of Bacterial Cell Suspension

To prepare bacterial cell suspensions, the protocol of Zhou et al. (2011) [16] was followed. Briefly, individually purified Bacillus isolates kept in NA-slants were first added to nutrient broth (NB) medium (100 mL) and incubated at 28 °C for 24 h with rotary agitation (150 rpm). Afterward, centrifugation was performed at 5000 rpm for 5 min, and the cells were harvested by discarding the supernatant. The pellets were further washed and resuspended in sterile distilled water to obtain an initial bacterial population density of 1 × 10^9^ colony-forming units (CFU/mL) by adjusting at 640 nm using a spectrophotometer (double beam UV-VIS, Systronics, Gujrat, India). Finally, the suspension was diluted using the serial dilution procedure from 1 × 10^9^ to 1 × 10^5^, 1 × 10^6^, 1 × 10^7^, and 1 × 10^8^ CFU/mL.

### 2.3. Collection and Rearing of Red Spider Mite

The tested pest, Red Spider Mites (RSM) were collected from the Experimental Organic Tea Garden (26°43′17″ N, 94°11′49″ E) of Assam Agricultural University, Jorhat, Assam (India). A stock culture of RSM population was maintained by following the detached leaf culture technique [17]. The collected mites were immediately transferred onto fresh leaves of susceptible tea cultivar, TV1, keeping the leaves on moistened cotton pads (ca. 1.5 cm thick)placed in plastic rearing trays (42 × 30 × 6.5 cm). Withered leaves were replaced regularly with new ones at 4-day intervals. The rearing trays were kept under controlled conditions in the laboratory at a temperature of 27 ± 2 °C, 75–80% relative humidity, and 16L: 8D photoperiod (Appendix A). Periodically, water was added to rearing trays to keep the cotton moist and prevent further drying of the leaves.

### 2.4. Screening of Miticidal (Acaricidal and Ovicidal) Potential of Bacillus spp. against O. coffeae

Adulticidal and ovicidal activities of the three *Bacillus* isolates at three different concentrations (1 × 10^7^, 1 × 10^8^ and 1 × 10^9^ CFU/mL) were evaluated against *O. coffeae* in order to determine their pesticidal efficiency. The leaf disc method proposed by Ebeling and Pence (1953) [18] was adopted for both the bioassays at a temperature of 27 ± 2 °C, 75–80% relative humidity and 16L: 8D photoperiod [17].

### 2.5. Setup for Adulticidal Activity

The matured tea leaves (var. TV1) with no past history of pesticidal application for 15 days were collected from the Organic Tea Garden of Assam Agricultural University, Jorhat, Assam (India). Leaf discs (20 mm diameter cut from these mature tea leaves) were placed in Petri dishes lined with water-saturated cotton wool. With a fine camel-hair brush, 10 adult female mites were carefully introduced onto the surface of the discs and counted under a binocular microscope. To evaluate the efficacy of different *Bacillus* spp. against adult *O. coffeae*, direct spraying of bacterial isolates was done at three concentrations of 1 × 10^7^, 1 × 10^8^, and 1 × 10^9^ CFU/mL, adding 4 drops of Tween-20 as an emulsifier. Distilled-water-sprayed leaf discs served as control. Each Petri dish containing leaf discs was sprayed with a uniform quantity (1.2 mL/5 s) of *Bacillus* spp., using a manual glass atomizer (50 mL). The surviving mites were counted at 24 h intervals up to 96 h. Each experiment was under taken with six replications.

The mortality percentage was calculated and corrected using Abbott’s formula [19].
Corrected mortality (%) = Mortality in treatment (%) − Mortality in control (%)/100 − Mortality in control (%) × 100.

### 2.6. Setup for Ovicidal Activity

As many ten gravid female mites were introduced onto the surface of the discs (24 h before the start of the experiment), allowed to lay eggs, and finally, the number of eggs was adjusted to 30 eggs/leaf disc. The leaf discs containing the eggs were sprayed with different *Bacillus* isolates at 1 × 10^7^ to 1 × 10^9^ CFU/mL, adding 4 drops of Tween-20, using a glass atomizer (constant pressure 2.5 kg/cm^2^) and distilled water served as the control. The discs were dried for at least 30 min. After drying, the discs were placed at 27 ± 2 °C and 65 ± 5% relative humidity. All discs were examined on a daily basis for 14 successive days. Hatchability was recorded for both experimental and control batches of eggs. The eggs failing to hatch after this period were considered non-viable. Six replicates were used for each treatment [20].

### 2.7. Virulence of the Three Bacillus Isolates against Adults and Eggs of O. Coffeae

The virulence of all three strains of *Bacillus* was evaluated to estimate LC_50_ following the aforementioned leaf disc method [18]. All three strains showed high pathogenicity in the preliminary screening assay and were further tested against both adults and eggs of RSM at 1 × 10^5^, 1 × 10^6^, 1 × 10^7^, 1 × 10^8^, and 1 × 10^9^ CFU/mL to find out its virulency. For each concentration, six replications were performed. Adult mortality was recorded at every 24 h interval up to 96 h, and in the case of ovicidal activity, data were recorded up to 14 successive days. And subsequently, median lethal concentration (LC_50_) of all the three treatments to adults and eggs of *O. coffeae* was calculated.

### 2.8. Electron Microscopy Study

The virulence of the most effective *Bacillus* isolate against adults and eggs of *O. coffeae* was studied with digital bright field microscopy under 40× (Carl Zeiss, Axio Lab 5, Jena, Germany) and was further confirmed by Scanning Electron Microscopic study. Samples of dead mites and unhatched eggs of *O. coffeae* were prepared for SEM study as per protocol suggested by Orion et al. (1994) [21] with slight modifications. Briefly, the egg samples were fixed at 4 °C in closed tubes containing 1.25% glutaraldehyde and 1.25% paraformaldehyde in 0.05 M cacodylate buffer (pH 7.2) for 4 h. The samples were treated serially for 15 min with 50, 70, 90, and 100% ethanol and butanol mixture. The samples were then vacuum freeze-dried for 24 h. The samples were coated by carbon grid with platinum coating and observed under JEOL (Tokyo, Japan), JSM- 6390LV for morphological changes in insect bodies and egg surface using SAIC Facility at Tezpur University, Tezpur, Assam (India).

### 2.9. Characterization of Pesticidal Metabolites through LC-MS Profiling

The presence of secondary metabolites with pesticidal properties in the supernatants of three *Bacillus* isolates was determined by liquid chromatography mass spectrometry (LC-MS) analysis at CSIR-Central Drug Research Institute (CDRI, Uttar Pradesh, India), Lucknow (India). The *Bacillus* strains were allowed to grow in Nutrient Broth (NB) (Sigma-Aldrich, St. Louis, MO, USA) in conical flasks (500 mL), placed in a rotary shaker (Lark Innovata, Germany) at 120 rpm for 4 days, and incubated at 28 ± 2 °C. The media was then filtered through Whatman No.1 filter paper and kept in another conical flask (500 mL). The filtrates were then mixed with an equal volume of methanol and twice the volume of chloroform (Methanol:Chloroform: 1:2) and kept at 28 ± 1 °C overnight. The solvents were further centrifuged at 12,000 rpm for 10 min and the supernatant obtained was dried in a rotary evaporator system (IKA^®^, Staufen, Germany). The powdered extract of *Bacillus* isolates obtained was re-dissolved in methanol. The extract was filtered through 0.22 µ syringe filter and outsourced for performing LC-MS at CSIR-CDRI (India). The LC-MS peaks were analyzed in the MestreNova (Mnova Suite V.11.0.4) software against known insecticidal compounds listed in the literature [22]. The respective retention time and similarity score were recorded for each compound.

### 2.10. Statistical Analysis

Data on mortality and ovicidal activities of *Bacillus* cell suspension against *O. coffeae* were generated, arcsine transformed, and subjected to analysis of variance (Tukey’s test of significance at 5% level) using SPSS version 20.0 [23].

## 3. Results

### 3.1. In Vitro Adulticidal Activity of Bacillus spp. against O. Coffeae

All the evaluated *Bacillus* spp., *B. amyloliquefaciens* BAC1, *B. subtilis* LB22, and *Bacillus velezensis* AB22, expressed both adulticidal (Figure 1) and ovicidal activity to varying proportions at the three different concentrations tested (1 × 10^7^, 1 × 10^8^, and 1 × 10^9^ CFU/mL).

At 1 × 10^9^ CFU/mL, the isolate *B. velezensis* AB22 showed 100% adult mortality earliest at 72 h after treatment (HAT), making it the most efficient among the other treatments. Another isolate, *B. amyloliquefaciens* BAC1 at the same concentration, also showed 100% mortality of adult *O. coffeae* at 96 HAT, which was found to be statistically at par with *B. velezensis* AB22, showing 96.66% mortality at 1 × 10^8^ CFU/mL, followed by *B. subtilis* LB22 with 93.33% mortality at 1 × 10^9^ CFU/mL (Appendix A).

### 3.2. In Vitro Ovicidal Activity of Bacillus spp. against O. coffeae

As shown in Figure 2 and Figure 3C,D, the three isolated *Bacillus* spp. showed varying degrees of egg mortality after 14 days after spray (DAS) at all three concentrations tested. The isolate *B. velezensis* AB22 showed 100% ovicidal activity of *O. coffeae* at 1 × 10^9^ CFU/mL followed by *B. amyloliquefaciens* BAC1 (92.22%) and *B. subtilis* LB22 (86.67%). The results further indicated that higher concentrations of *Bacillus* isolates were associated with proportionately higher pesticidal efficacy against the insect.

The bio-efficacy of *B. velezensis* AB22 showed higher adulticidal and ovicidal activity against *O. coffeae*. The *Bacillus* spp., *B. velezensis* registered a net 6.00% higher ovicidal activity over the other two *Bacillus* spp. (*B*. *amyloliquefaciens* and *B. subtilis*). However, the ovicidal activity of all three *Bacillus* spp. was observed to be significantly higher by comparing the response at 1 × 10^7^ versus 1 × 10^9^ CFU/mL (Appendix A).

### 3.3. Virulence of the Three Bacillus Isolates against Adults and Eggs of O. Coffeae

The median lethal concentration (LC_50_) of all three *Bacillus* spp. was measured for their efficacy against the mortality of adults and eggs of *O. coffeae*. The LC_50_ values for adult mortality by *Bacillus velezensis* AB22, *B. amyloliquefaciens* BAC1 and *B. subtilis* LB22 were 0.28 × 10^5^, 1.06 × 10^5^ and 5.12 × 10^7^ CFU/mL, respectively (Table 1). In case of egg mortality, LC_50_ values were found to be in the same trend i.e., 0.29 × 10^5^, 0.41 × 10^5^ and 0.43 × 10^7^ CFU/mL for *Bacillus velezensis* AB22, *B. amyloliquefaciens* BAC1 and *B. subtilis* LB22 (Table 2). Thus, *Bacillus velezensis* AB22, showing the lowest LC_50_ values for both adults and eggs of *O. coffeae*, i.e., 0.28 × 10^5^ and 0.29 × 10^5^,respectively, was considered the most virulent isolate amongst the tested *Bacillus* spp.

### 3.4. Morphological Changes in O. coffeae in Response to Bacillus Treatment

SEM study on morphological changes in *O. coffeae* adults and eggs in response to the most effective *Bacillus* isolate, *B. velezensis* AB22 treatment showed complete paralysis of adult mites with distorted abdomen and folded legs at 96 h after treatment (HAT). The inner body content of the mite dried completely and remained as a mass of cast skin (Figure 4A). The eggs of *O. coffeae* after treatment (14 DAS) were also disfigured, flattened on one side, and remained unhatched (Figure 4B).

### 3.5. Profiling and Identification of Pesticidal Metabolites of Bacillus spp.

*Bacillus* species are known to secrete various metabolites inhibitory to many phytopathogens. Our study identified an array of insect-pests attacking diversified host plants. The LC-MS chromatogram of extracts from three *Bacillus* isolates depicted several peaks corresponding to bioactive compounds recognized by their peak retention time, peak area (%), height (%), and mass spectral fragmentation patterns of known compounds described by the National Institute of Standards and Technology (NIST, Gaithersburg, MD, USA) library. The presence of different compounds was observed to have insecticidal as well acaricidal activities. These compounds comprised brevianamide A, heptadecanoic acid, thiolutin, and versimide. In addition to these four metabolites, *B. velezensis* was observed to produce citromycin, emodin, peramine, and zwittermicin A; while *B. subtilis* was characterized by some unique compounds, namely, anhydrofusarubin, emodin, nikkomycins, and sterigmatocystin. On the contrary, *B. amyloliquefaciens* produced tenuazonic acid and milbemycins D as additional metabolites (Table 3, Table 4 and Table 5 and Figure 5, Figure 6 and Figure 7).

## 4. Discussion

The target pest, *Oligonychus coffeae*, is one of the major mite pests causing a serious threat to tea crops, deteriorating both production as well as quality [5]. Microorganisms, such as antagonistic fungi and bacteria, provide a much safer, more sustainable and environmentally friendly alternative to the commercially available synthetic acaricides [24,25,26]. Numerous investigations revealed that certain *Bacillus* species have pesticidal effects against various diseases and insect pests in a wide range of crops [14,27,28,29,30]. In this investigation, we employed three rhizospheric *Bacillus* spp. (*B. amyloliquiefaciens* BAC1, *B. subtilis* LB22 and *B. velezensis* AB22) to explore their miticidal potential, if any. Our study demonstrated the adulticidal and ovicidal efficacy of all three *Bacillus* spp. Against *O. coffeae* under laboratory conditions at three different concentrations:1 × 10^7^, 1 × 10^8^, and 1 × 10^9^ CFU/mL. However, the highest efficacy on adults and eggs of mites was recorded in *B. velezensis* AB22 (1 × 10^9^ CFU/mL), followed by *B. amyloliquefaciens* BAC1 (1 × 10^9^ CFU/mL), and *B. subtilis* LB22 (1 × 10^9^ CFU/mL) (Appendix A). *B. velezensis* AB22, with a lower LC_50_ value of 0.28 × 10^5^, was found superior to the other two *Bacillus* isolates, *B. amyloliquefaciens* BAC1 and *B. subtilis* LB22, recording LC_50_ values of 1.06 × 10^5^ and 5.12 × 10^7^, respectively, after 96 HAT. Further, histopathological studies of the adults and eggs of *O. coffeae* treated with the most effective strain, i.e., *B. velezensis*, showed complete paralysis of adult mites with distorted abdomen and folded legs at 96 h after treatment when observed under compound and Scanning Electron Microscope. Additionally, the treated eggs of *O. coffeae* showed disfiguration, flattened on one side, and remained unhatched at 14 days after treatment. This may be attributed to the pesticidal compounds, namely, microbial lipopeptides and extracellular cuticle-degrading enzymes, including chitinases and proteases documented in *Bacillus* species [31]. The cuticle-degrading hydrolytic enzymes can effectively degrade the chitin and protein components in the cuticle, which impairs the exoskeleton’s essential function in the life of pests, leading to lethal effects and significantly decreasing the ecological fitness of the surviving insects [31].

The pesticidal efficacy of the genus *Bacillus* has been documented in some previous studies, showing bioefficacy against various insect pests. Choi et al. (2023) [27] tested the crude enzymes and bacterial broth culture of *Bacillus velezensis* CE 100 against gall midge larvae and found that both treatments caused mortality rates of up to 85.5% and 96.7%, respectively. *B. amyloliquefaciens* strains QST2808, FZB25, D747, and QST 2808 were found to exhibit high mortality of adults and eggs of spider mites in pepper [29] and aphicidal effect against bean aphids via treatment with a suspension of bacterial cells (CS), cell-free supernatant (CFS), and isolated lipopeptide fraction (LF) of strains CBMDDrag3, PGPBacCA2, and CBMDLO3 on artificial diet [30] were reported. Al-Azzazy et al. (2020) [32] also tested the effectiveness of *Bacillus subtilis* (2.470 × 10^8^ CFU/mL) and *Bacillus qassimus* (3.320 × 10^8^ CFU/mL) against *Tetranychus urticae,* infesting eggplant, under laboratory conditions and found that seven days post-treatment with *B. subtilis* and *B. qassimus* caused 72.22 and 70.74% reductions in mite populations, respectively. Similarly, at 96 h after exposure, a *Bacillus thuringiensis* strain was reported to have median lethal doses (LC_50_) against larvae of *Helicoverpa armigera* ranging from 1.7 to 1.8 × 10^5^ CFU/mL [33].

Besides the well studied cuticle degrading enzymes and proteins, some more pesticidal secondary metabolites may also contribute in the acaricidal efficacy of the *Bacillus* spp. Our study through LC-MS profiling of putative pesticidal compounds of the three *Bacillus* isolates identified a total number of nine (9) metabolites in *Bacillus velezensis* AB22, six (6) in *Bacillus amyloliquifaciens* BAC1 and eight (8) in *Bacillus subtilis* LB22. Four proven insecticidal metabolites, Brevianamide A [34], Heptadecanoic acid [31], Thiolutin [35], and Versimide [36], were also observed in all three studied *Bacillus* spp. Two identified secondary metabolites, sterigmatocystin and milbemycins D reported from *B. subtilis* and *B. amyloliquefaciens*, respectively, have proven acaricidal properties. Mishima (1983) [37] reported high potency of milbemycins D against a variety of mites supporting our finding, while sterigmatocystin is reported to be highly toxic to mold mites, *Tyrophagus putrescentiae* [38]. From all the experiments carried out in our study, the isolate *Bacillus velezensis* AB22 was found to be superior to the other two *Bacillus* species. This may be due to the presence of identified pesticidal compounds compared to the other two isolates or due to the secretion of Zwittermicin, which has been widely reported antibacterial and antifungal compound [39] and is also known for its insecticidal action as well-known entomopathogenic bacteria. Zwittermicin A enhancing the activity of endotoxin from *B. thuringiensis* [40,41] is reported to increase the mortality of third instar gypsy moth, *Lymantria dispar* (L) treatment with *B. thuringiensis* [42]. Hence, the present investigation established the miticidal (adulticidal and ovicidal) property of all three *Bacillus* isolates with varying efficacy, reporting such findings for the first time against *O. coffeae* of tea to the best of our knowledge.

The present bioassay, coupled with the characterization of key pesticidal compounds, clearly demonstrated the untapped acaricidal and entomopathogenic potential of *Bacillus* spp. However, it further calls for a much greater depth of study on decoding molecular and biochemical principles of tripartite interaction involving bioagent tea as host plant mite pest followed by isolation of key functional metabolites for robust field management of *O. coffeae*. These studies also put forth an additional possibility of biocloning secondary metabolites as synthetic microbes.

## 5. Conclusions

In our work, we unraveled the miticidal potential of rhizospheric *Bacillus* spp. (*B. amyloliquiefaciens, B. subtilis*, and *B. velezensis*) against *O. coffeae* of tea, with the highest efficacy of *B. velezensis,* and the putative metabolites employed by all the bacterial strains against the pest. Light microscopy coupled with SEM study further revealed the morphological deformities due to the most efficient strain *B. velezensis* BAC1 in adults and eggs of *O. coffeae*. The study, with the initial clue on the pesticidal efficacy of *Bacillus* spp., can open up a new frontier in harnessing these PGP bacterial species (unpublished data) also as a biocontrol option against mite pests. However, further investigation is required for in planta acaridical efficacy of the strains under field conditions in *O. coffeae* and in other pests as well. Furthermore, purification and screening of individual pesticidal compounds can be a new-generation technology alternative to chemical acaricides in tea.

## Figures and Tables

**Figure 1 microorganisms-11-02691-f001:**
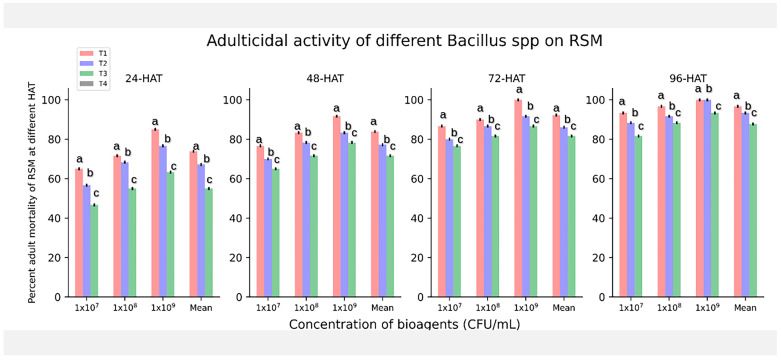
Effect of different *Bacillus* spp. at 1 × 10^9^ CFU/mL on *O. coffeae* adults at 24-hourly interval for 96 h of treatment. T_1_: *Bacillus velezensis* AB22; T_2_: *B. amyloliquefaciens* BAC1; T_3_: *B. subtilis* LB22; T_4_: Control. Different lower-case letters (a, b, and c) show significant differences (*p* = 0.05) between treatments.

**Figure 2 microorganisms-11-02691-f002:**
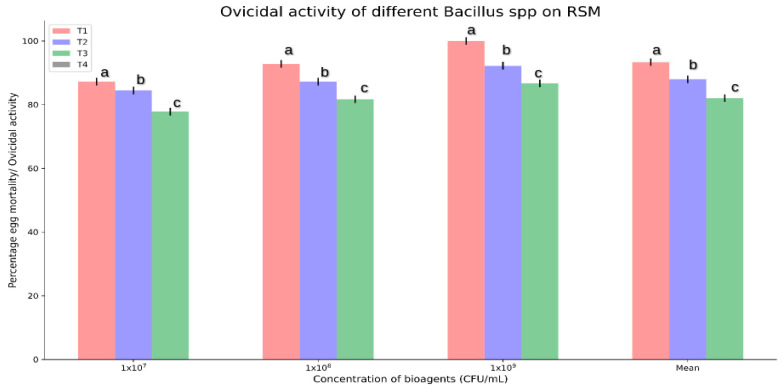
Ovicidal effect of *Bacillus* spp. at three different population doses (1 × 10^7^–1 × 10^9^ CFU/mL) against *O. coffeae* during 14 days of treatments. T1: *Bacillus velezensis* AB22; T2: *B. amyloliquefaciens* Bac1; T3: *B. subtilis* LB22; T4: Control. Different lower-case letters (a, b, and c) show significant differences (*p* = 0.05) between treatments.

**Figure 3 microorganisms-11-02691-f003:**
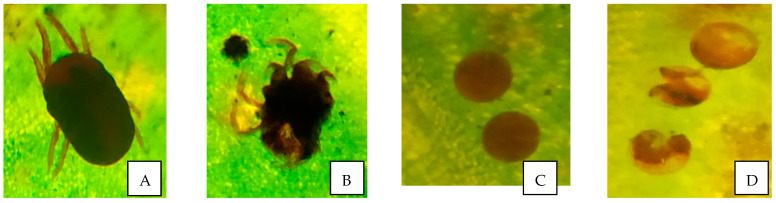
Photographs showing adulticidal and ovicidal activity of *Bacillus velezensis* AB22 against *O. coffeae* under 40×. (**A**) Adult mite in control (96 HAT); (**B**) Distorted adult mite (96 HAT); (**C**) Eggs of mite in control (14 DAS); (**D**) Damaged mite eggs (14 DAS).

**Figure 4 microorganisms-11-02691-f004:**
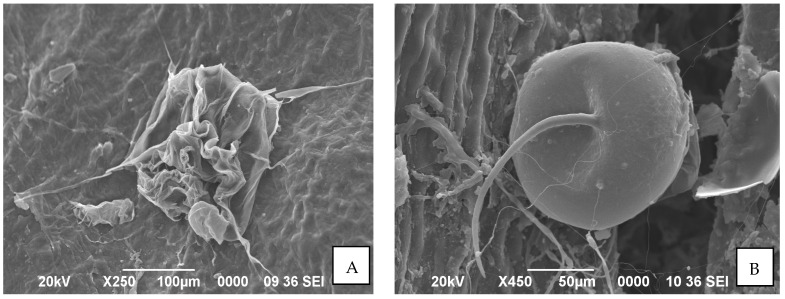
Morphological changes in *O. coffeae* in response to treatment with *Bacillus velezensis* AB22 under Scanning Electron Microscope. (**A**) Morphological distortion in treated adults; (**B**) Morphological distortion in treated eggs.

**Figure 5 microorganisms-11-02691-f005:**
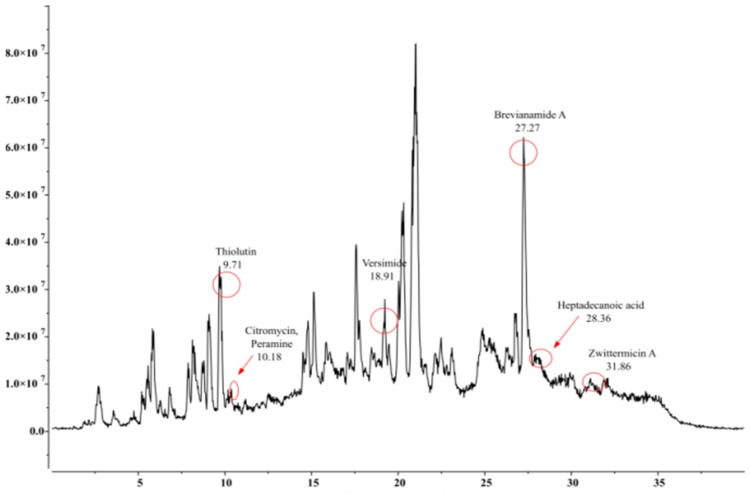
Detection of insecticidal metabolites by LC-MS from methanol crude extract of *Bacillus velezensis* AB22 (ON209629).

**Figure 6 microorganisms-11-02691-f006:**
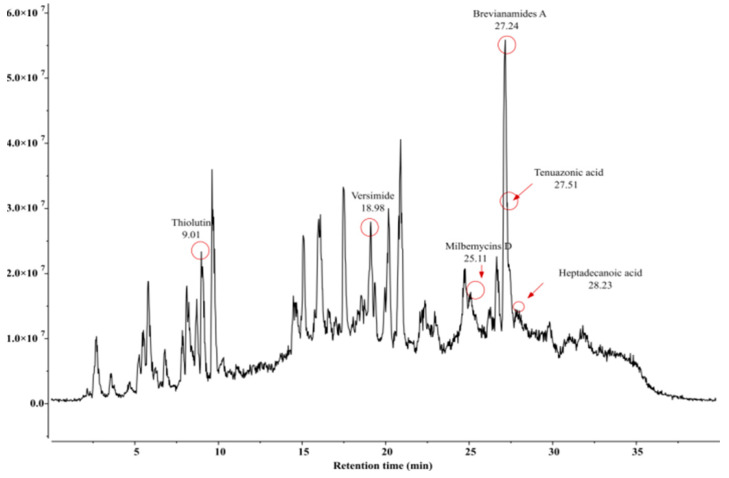
Detection of insecticidal metabolites by LC-MS from methanol crude extract of *Bacillus amyloliquifaciens* BAC1 (ON392425).

**Figure 7 microorganisms-11-02691-f007:**
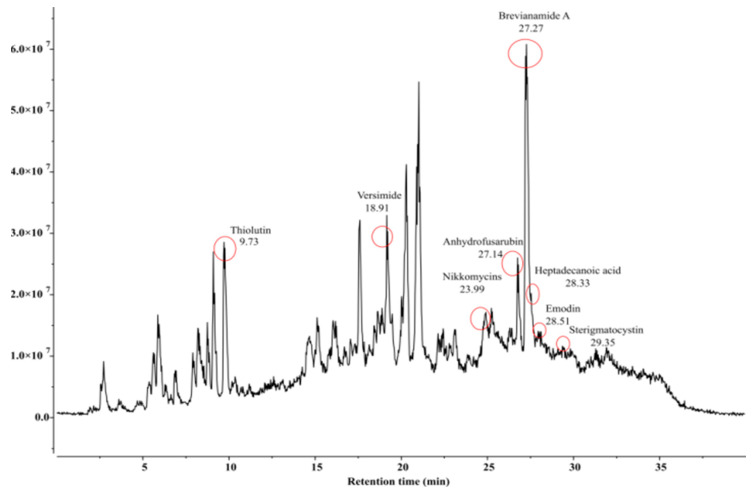
Detection of insecticidal metabolites by LC-MS from methanol crude extract of *Bacillus subtilis* LB22 (ON386193).

**Table 1 microorganisms-11-02691-t001:** LC_50_ of *Bacillus* isolates against adults of *O. coffeae* using different concentrations (1 × 10^5^ to 1 × 10^9^ CFU/mL) after 96 h of treatment.

Treatment	Conc. (CFU/mL)	Adult Mortality (%) after 96 h	Log of Conc.	Probit Mortality	Regression Statistics (a = Slope, b = Intercept)	Regression Equation Y = aX + b	LC_50_ (Y = 5) LC_50_ = antilogX	LC_50_ Value
**T1: *Bacillus velezensis* AB22**	1 × 10^5^	70.00 (56.79)	5	5.18	a = 0.262; b =3.832	Y = 0.262x + 3.832	28,708.79737	0.28 × 10^5^
	1 × 10^6^	81.67 (64.60)	6	5.39				
	1 × 10^7^	93.33 (75.00)	7	5.67				
	1 × 10^8^	96.67 (79.37)	8	5.81				
	1 × 10^9^	100.00 (90.00)	9	6.28				
**T2: *B. amyloliquefaciens* BAC1**	1 × 10^5^	63.33 (52.71)	5	5.08	a = 0.273; b = 3.643	Y = 0.273x + 3.643	106,151.4425	1.06 × 10^5^
	1 × 10^6^	76.67 (61.07)	6	5.28				
	1 × 10^7^	88.33 (70.00)	7	5.52				
	1 × 10^8^	91.67 (73.15)	8	5.61				
	1 × 10^9^	100.00 (90.00)	9	6.28				
**T3: *B. subtilis* LB22**	1 × 10^5^	61.67 (51.71)	5	5.05	a = 0.161; b = 4.229	Y = 0.161x + 4.229	51,286,138.4	5.12 × 10^7^
	1 × 10^6^	68.33 (55.73)	6	5.15				
	1 × 10^7^	81.67 (64.60)	7	5.39				
	1 × 10^8^	88.33 (70.00)	8	5.52				
	1 × 10^9^	93.33 (75.00)	9	5.67				

Y = Probit kill; X = Log concentration.

**Table 2 microorganisms-11-02691-t002:** LC_50_ of the *Bacillus* spp. most effective against eggs of *O. coffeae* using different concentrations (1 × 10^5^ to 1 × 10^9^ CFU/mL) after 14 days of treatment.

Treatment	Conc. (CFU/mL)	Ovicidal Activity (%) after 14 Days	Log of Conc.	Probit Mortality	Regression Statistics (a = Slope, b = Intercept)	Regression Equation Y = aX + b	LC_50_ (Y = 5) LC_50_ = antilogX	LC_50_ Value
**T1: *Bacillus velezensis* AB22**	1 × 10^5^	73.33 (58.89)	5	5.61	a = 0.238; b =3.936	Y= 0.238x + 3.936	29,552.09	0.29 × 10^5^
	1 × 10^6^	81.11 (64.23)	6	5.88				
	1 × 10^7^	87.22 (69.06)	7	6.13				
	1 × 10^8^	92.78 (74.41)	8	6.48				
	1 × 10^9^	100.00 (90.00)	9	8.95				
**T2: *B. amyloliquefaciens* BAC1**	1 × 10^5^	63.89 (53.01)	5	5.36	a = 0.130; b = 4.400	Y = 0.130x + 4.40	41,246.26	0.41 × 10^5^
	1 × 10^6^	76.67 (61.07)	6	5.74				
	1 × 10^7^	84.44 (66.77)	7	5.99				
	1 × 10^8^	87.22 (69.06)	8	6.13				
	1 × 10^9^	92.22 (73.81)	9	6.41				
**T3: *B. subtilis* LB22**	1 × 10^5^	61.11 (51.41)	5	5.28	a = 0.110; b = 4.490	Y = 0.110x + 4.49	43,287.61	0.43 × 10^5^
	1 × 10^6^	71.67 (57.80)	6	5.58				
	1 × 10^7^	77.78 (61.87)	7	5.77				
	1 × 10^8^	81.67 (64.65)	8	5.92				
	1 × 10^9^	86.67 (68.58)	9	6.13				

Y = Probit kill; X = Log concentration.

**Table 3 microorganisms-11-02691-t003:** Details of secondary metabolites (based on ECI/LC-MS) produced by *Bacillus velezensis* AB22 (ON209629).

Sl. No.	Compound Name	Formula	Match Score	Retention Time (RT)	Predicted *m/z*	Matched *m/z*
1.	Brevianamide A	C_21_H_23_N_3_O_3_	0.961	27.27	404.1371	403.8991
2.	Citromycin	C_13_H_10_O_5_	0.909	10.18	285.0160	284.8307
3.	Emodin	C_15_H_10_O_5_	0.912	28.51	303.0863	302.8130
4.	Heptadecanoic acid	C_17_H_34_O_2_	0.953	28.36	303.2894	302.9980
5.	Paxillines	C_27_H_33_NO_4_	0.872	12.46	458.2302	458.0682
6.	Peramine	C_12_H_17_N_5_O	0.905	10.18	285.1202	284.8307
7.	Thiolutin	C_8_H_8_N_2_O_2_S_2_	0.927	9.71	246.0365	245.7579
8.	Versimide	C_9_H_11_NO_4_	0.926	18.91	220.0580	219.8944
9.	Zwittermicin A	C_13_H_28_N_6_O_8_	0.856	31.86	414.2307	413.9633

**Table 4 microorganisms-11-02691-t004:** Details of secondary metabolites (based on ECI/LC-MS) produced by *Bacillus amyloliquifaciens* BAC1 (ON392425).

Sl. No.	Compound Name	Formula	Match Score	Retention Time (RT)	Predicted *m/z*	Matched *m/z*
1.	Brevianamides A	C_21_H_23_N_3_O_3_	0.933	27.24	404.1371	404.0471
2.	Heptadecanoic acid	C_17_H_34_O_2_	0.904	28.23	303.2894	302.9980
3.	Milbemycins D	C_33_H_48_O_7_	0.888	25.11	574.3738	574.1025
4.	Tenuazonic acid	C_8_H_16_O_2_	0.956	27.51	485.9953	486.2257
5.	Thiolutin	C_8_H_8_N_2_O_2_S_2_	0.970	9.01	246.0365	245.7579
6.	Versimide	C_9_H_11_NO_4_	0.923	18.98	220.0580	219.7834

**Table 5 microorganisms-11-02691-t005:** Details of secondary metabolites (based on ECI/LC-MS) produced by *Bacillus subtilis* LB22 (ON386193).

Sl. No.	Compound Name	Formula	Match Score	Retention Time (RT)	Predicted *m/z*	Matched *m/z*
1.	Anhydrofusarubin	C_15_H_12_O_6_	0.955	27.14	306.0972	305.9951
2.	Brevianamide A	C_21_H_23_N_3_O_3_	0.961	27.27	404.1371	403.8991
3.	Emodin	C_15_H_10_O_5_	0.912	28.51	303.0863	302.8130
4.	Heptadecanoic acid	C_17_H_34_O_2_	0.906	28.33	303.2894	303.0721
5.	Nikkomycins	C_20_H_25_N_5_O_10_	0.925	23.99	518.1494	518.3794
6.	Sterigmatocystin	C_18_H_12_O_6_	0.878	29.35	342.0972	341.8858
7.	Thiolutin	C_8_H_8_N_2_O_2_S_2_	0.982	9.73	246.0365	245.7949
8.	Versimide	C_9_H_11_NO_4_	0.926	18.91	220.0580	219.8944

## Data Availability

Data are contained within the article.

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
