# Peer review of "Rhizospheric Bacillus spp. Exhibit Miticidal Efficacy against Oligonychus coffeae (Acari: Tetranychidae) of Tea"

_microorganisms, 2023, doi:10.3390/microorganisms11112691_

Round 1
Reviewer 1 Report
Manuscript entitles “Rhizospheric Bacillus spp. exhibit miticidal efficacy against Oligonychus coffeae (Acari: Tetranychidae) of Tea” written by Bora et al., explaining the miticidal potential of Bacillus spp. against tea red spider mite the most serious pest of tea. The manuscript is well written, with some interesting information, and having the potential of reader’s interest. I have some suggestions which authors need to be address.
In title, italic the scientific names.
Line 12; to control this devastating pest……….
Line 13; area wise…………
Line 14; delete inhabited
Line 15; adulticidal and ovicidal activities. And delete the “against the pest”.
Line 56; Different Bacillus spp. have been reported to be remove the bold format.
Line 78; I cannot find the Figure S1
Line 83; write about the recipe of NA medium?
How the bacterial strains were stored? With or without glycerol? In NA broth or just on NA medium plates? Mostly stock solution of 50% (v/v) of glycerol is used to preserve bacteria at -80℃, and slants are used for fungi.
Line 89; update the reference the reference is too old; it is recommended to use last 20 years references.
Line 96; How the CFU was adjusted?
Where Is Figure S2?
Line 113; update the reference
Line 128; What is abbotts formula? Write it.
Line 143; Bacillus, make it italic
Line 156; same as above update the references
Line 179; in statistical analysis it is recommended to write about test name, which authors used to calculate significant difference among treatments.
Citation of Figure 2 is missing in the text.
Table 1 and 2 is not cited in the text and also not available in the manuscript body.
Figure 1, 2, 5, 6, and 7 quality is not good, the x-axis and y-axis are not very clear, it is recommended to provide a high-resolution figure.
Figure 4; where is the SEM figure for control?
Why SEM is not done for the other strains of Bacillus?
Discussion first paragraph is looks like a part of introduction. Authors just explained the results of previous studies but cannot compare their own results with the findings of previous studies. However, discussion section is written very poorly, authors need to rewrite the discussion section and need to discuss their all results in the light of previous findings. A lot of results are not discussed in the discussion section.
Line 312; delete median lethal concentrations
Line 317; update the reference
Line 343; authors stated that the used biocontrol agents are PGP? But in the manuscript, they have not tested any PGP traits of strains and no experiment was conducted related to PGP. How they find that the strains have the ability of PGP?
Update the conclusion section according to your results, similar like abstract.
English of manuscript need to be check, for better understanding.
Reviewer 2 Report
Review
The manuscript titled “Rhizospheric Bacillus spp. exhibit miticidal efficacy against Oligonychus coffeae (Acari: Tetranychidae) of Tea” written byBora et al., reported about the biocontrol efficacy of three different Bacillus spp. previously isolated from rhizospheric soil of agricultural fields against O. coffeae using their adulticidal and ovicidal properties under controlled laboratory conditions. Authors also studied identification of effective metabolites responsible for their pesticidal activity to develop a novel approach against mite pests for organic tea ecosystem.
Authors studied adulticidal and ovicidal activities of the three Bacillus isolates at three different concentrations comparing them with each other. However it is difficult to estimate efficiency of studied bacilli without comparison with the effect of chemical or biological acaricide used in practice.
Remarks
Title
Row 2. Use italics for Bacillus and Oligonychus coffeae
Abstract
Row 14,23,25,60,201,331,338,343. spp should be with dot
Material and methods
Row 86. Bacillus isolate.. should be Bacillus spp. isolates
Row 143. Use italics for Bacillus and genus before Bacillus
Row 146,218. adults? Correct CFU/ml to mL
Row 168. rotary shaker- model, company, country?
Results
Row 218,223,234. adults?
Discussion
Row 308. Use italics for Bacillus
Row 318,323,326.Correct B. velenzensis.
Conclusion
Row 345. Use italics for in planta
See attached file

Round 2
Reviewer 1 Report
Abbott’s formula is not clear, authors need to cross check it, authors must used (), or / to provide seperation between the statements.
Authors stated that statistical analysis was performed using Complete Randomized Design. I think authors have no idea about the statistical test. CRD is design under experiments were performed, while statistical test such as t-test, LSD, Duncans, Tucky tests are used to show the significant difference among treatments. At which p value author check a significant difference among treatments.
How the error bar were generated? how the data was represent at SEM or SED?
Figure 1 and Figure 2, its necessary to show the significant difference among treatments using a statistical test.
Author Response
Reviewers comments : Abbott’s formula is not clear, authors need to cross check it, authors must used (), or / to provide separation between the statements.
Response: We have cross-checked the Abbott’s formula and provided the corrected Abbott’s formula at line no 157-158 , sorry for inconvenience . We have already used (19) at line 155.
Reviewers comments : Authors stated that statistical analysis was performed using Complete Randomized Design. I think authors have no idea about the statistical test. CRD is design under experiments were performed, while statistical test such as t-test, LSD, Duncans, Tucky tests are used to show the significant difference among treatments. At which p value author check a significant difference among treatments.
Authors response : We have added the statistical test as Tukey test at 5% level of significance as mentioned through line 221-222 in our another revised version of manuscript.
Reviewers comments : How the error bar were generated? how the data was represent at SEM or SED?
Authors response: We have used SEM while generating error bar .
Reviewers comments : Figure 1 and Figure 2, its necessary to show the significant difference among treatments using a statistical test.
Authors response : Yes, we do agree with your excellent suggestion . accordingly , we have redrawn Fifure1 and Figure 2 carrying lower case letters showing significant differences between different treatments.
Thanks a lot for their all efforts to improve this manuscript.
(Popy Bora et al. , Assam Agricultural University , Jorhat , Assam , INDIA)
